# Emergent chirality in a polar meron to skyrmion phase transition

Yu-Tsun Shao [1,2], Sujit Das [3,4], Zijian Hong [5,6], Ruijuan Xu [7,8,9], Swathi Chandrika[1], Fernando Gómez-Ortiz [10], Pablo García-Fernández [10], Long-Qing Chen [5], Harold Y. Hwang [7,8], Javier Junquera [10], Lane W. Martin [3,11], Ramamoorthy Ramesh [4,11,12,13] & David A. Muller [1,14] ✉

Polar skyrmions are predicted to emerge from the interplay of elastic, electrostatic and gradient energies, in contrast to the key role of the antisymmetric Dzyalozhinskii-Moriya interaction in magnetic skyrmions. Here, we explore the reversible transition from a skyrmion state (topological charge of −1) to a two-dimensional, tetratic lattice of merons (with topological charge of −1/2) upon varying the temperature and elastic boundary conditions in $[(PbTiO_3)_{16}/(SrTiO_3)_{16}]_8$ membranes. This topological phase transition is accompanied by a change in chirality, from zero-net chirality (in meronic phase) to net-handedness (in skyrmionic phase). We show how scanning electron diffraction provides a robust measure of the local polarization simultaneously with the strain state at sub-nm resolution, while also directly mapping the chirality of each skyrmion. Using this, we demonstrate strain as a crucial order parameter to drive isotropic-to-anisotropic structural transitions of chiral polar skyrmions to non-chiral merons, validated with X-ray reciprocal space mapping and phase-field simulations.

A structural transition in materials involves the rearrangement of a periodic array of motifs in response to external stimuli, which governs the materials' functional properties. Such a transition is not limited to atoms, but also anticipated in lattices consisting of unconventional quasi-particles such as skyrmions or merons[1–4]. The recent discovery of non-trivial topological textures, including flux-closure[5–8], vortices[9–15], skyrmion[16–21], merons[22–24], or hopfions[25] in ferroelectric-oxide nanostructures provides a framework for exploring topology and exotic physical phenomena in condensed matter physics with a focus on polar order[26–28]. Polar skyrmion bubbles consist of three-dimensional (3D) electric dipole textures which, plane by plane, are characterized by an integer topological charge, called the skyrmion number, and defined as

$$N_{sk} = \frac{1}{4\pi} \iint d^2 r \, \vec{n} \cdot \left( \frac{\partial \vec{n}}{\partial x} \times \frac{\partial \vec{n}}{\partial y} \right), \qquad (1)$$

[1]School of Applied and Engineering Physics, Cornell University, Ithaca, NY, USA. [2]Mork Family Department of Chemical Engineering and Materials Science, University of Southern California, Los Angeles, CA, USA. [3]Department of Materials Science and Engineering, University of California, Berkeley, CA, USA. [4]Materials Research Centre, Indian Institute of Science, Bangalore, India. [5]Materials Research Institute and Department of Materials Science and Engineering, The Pennsylvania State University, University Park, PA, USA. [6]Laboratory of Dielectric Materials, School of Materials Science and Engineering, Zhejiang University, Hangzhou 310027, China. [7]Department of Applied Physics, Stanford University, Stanford, CA, USA. [8]Stanford Institute for Materials and Energy Sciences, SLAC National Accelerator Laboratory, Menlo Park, CA, USA. [9]Department of Materials Science and Engineering, North Carolina State University, Raleigh, NC, USA. [10]Departamento de Ciencias de la Tierra y Física de la Materia Condensada, Universidad de Cantabria, Cantabria Campus Internacional, Avenida de los Castros s/n, 39005 Santander, Spain. [11]Materials Sciences Division, Lawrence Berkeley National Laboratory, Berkeley, CA, USA. [12]Department of Physics, University of California, Berkeley, CA, USA. [13]Rice University, Houston, TX, USA. [14]Kavli Institute at Cornell for Nanoscale Science, Ithaca, NY, USA. ✉e-mail: david.a.muller@cornell.edu

where $\vec{n}$ is the normalized local dipole moment. Merons are another kind of topological solitons. To understand the differences in their topology, Fig. 1 sketches the dipolar textures for a Bloch-like skyrmion versus a meron. As shown in Fig. 1A, B, at the central $xy$-plane, a polar skyrmion exhibits an out-of-plane polarization ($P_{op}$) at the core and an antiparallel $P_{op}$ outside the boundary (magenta arrows, Fig. 1A, B). The polarization continuously rotates in such a way that the in-plane component shows a curling Bloch-like pattern[29] (blue/yellow arrows, Fig.1A, B) that provides chirality to the skyrmion[20]. In contrast, merons (Fig. 1C–E) exhibit $P_{op}$ at the core and gradually evolve to in-plane $P_{ip}$ at the periphery, without being fully surrounded by an antiparallel $P_{op}$ component[30]. As an example, a meron and a skyrmion can have the same $P_{ip}$ texture but vary only by their $P_{op}$ components. Therefore, simultaneous three-dimensional experimental characterization is necessary to correctly determine the topology of the polar textures, and we show how both in-plane and out-plane polar order can be mapped simultaneously.

A fundamental question pertaining to the spatial arrangement of the skyrmions is the degree of long-range order, if any, amongst the skyrmions, in terms of both the orientational order as well as translational order[4,31]. The emergence of long range order, or conversely the disappearance of a possible long range order in the polar skyrmion lattice driven by temperature can provide a heretofore unexplored possibility of a melting phase transition in such a topologically protected two-dimensional (2D) array of polar skyrmions (akin to the well-known Kosterlitz-Thouless-Halperin-Nelson-Young (KTHNY) transition)[4,32]. Although phase field models had predicted the possibility of forming a long-range ordered skyrmion lattice for certain values of mismatch strain with the substrate[33], experimentally this is yet to be demonstrated. Given that the magnitude and sign (compressive vs. tensile) of the strain in the superlattice is a critical component of such long-range order, we sought to manipulate this by lifting off the superlattice from the substrate[34–36]. In doing so, we are able to study the ground state of the skyrmions without any interference from substrate constraint. On such a free-standing membrane, we then imposed different elastic boundary conditions by varying the temperature, due to differences in thermal expansion of the two materials, to manipulate the degree of long-range order in the skyrmion lattice.

Here, we report the direct observation of sequential structural transformations for the polar skyrmions: from stripe-shaped to circular-shaped disordered skyrmions bubbles, to a tetratic-ordered meron lattice in lifted-off [(PbTiO$_3$)$_{16}$/(SrTiO$_3$)$_{16}$]$_8$ superlattice membranes through integrated experimental measurements and theoretical phase-field calculations. Electron imaging of the polar distortions can be challenging due to local sample mis-tilt artifacts which can affect or dominate the contrast. To disentangle tilt and polarization, we developed an approach for the analysis of Kikuchi bands by using the intensity asymmetry in diffuse scattering, recorded by four-dimensional scanning transmission electron microscopy (4D-STEM) with an electron microscopy pixel array detector (EMPAD)[37]. Using the high-dynamic-range EMPAD, we were able to collect the full scattering distribution, including Kikuchi bands and Bragg reflections (>10$^4$ difference in diffracted intensities), which also contains lattice information for determining the strain state of the specimen. Using 4D-STEM and dynamical diffraction analysis we also demonstrate, for the first time, a direct experimental determination of chirality of individual polar skyrmions. Our observations reveal the emergence of a square meron lattice with $N_{sk} = -1/2$ from the disordered skyrmion phase with a $N_{sk} = -1$, in which the chirality changed from left-handed to zero-net chirality, respectively.

## Results

### Experimental setup of 4D-STEM

To examine the local polarization distribution, we performed 4D-STEM experiments on plan-view, lifted-off samples (Fig. S1) to image the in-plane Bloch components (details in Methods). Briefly, 4D-

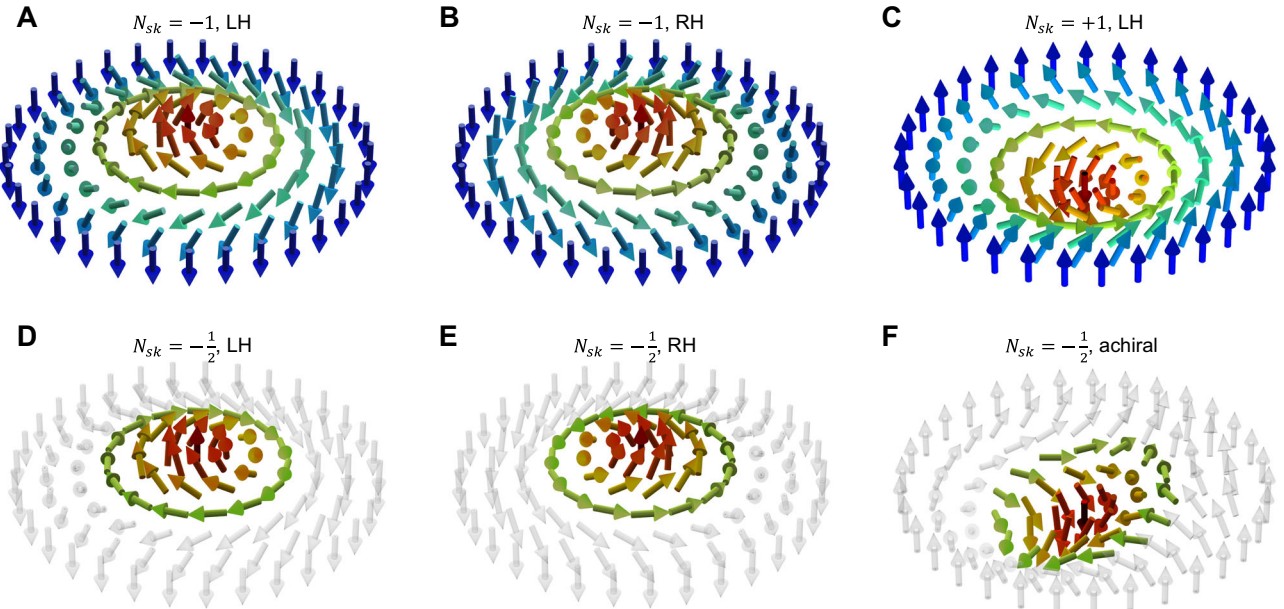

**Fig. 1 | Sketches of the dipolar textures observed for merons versus skyrmions.** Schematic of Bloch-like skyrmions with (**A**) right-handed and (**B**) left-handed chiralities, both possessing a topological charge of $N_{sk} = -1$. A common feature of both the Bloch ($\nabla \times \boldsymbol{P} \neq 0, N_{sk} = -1$) and Néel ($\nabla \cdot \boldsymbol{P} \neq 0, N_{sk} = -1$) skyrmions is the out-of-plane polarization rotates from maximal at the core, to maximal in the opposite direction at the periphery. Instead, the meronic phases can be characterized by a vanishing out-of-plane polarization at the periphery, and a non-zero value at the core. The in-plane component of the polarization presents a non-vanishing vorticity. In the case of a vortex, both right-handed (**C**), and left-handed (**D**) chiralities are possible, with a topological charge of $N_{sk} = -1/2$. In the case of an antivortex, and although the total topological charge might remain invariant if we reverse the direction of the out-of-plane polarization (**E**), the final meron configuration is achiral.

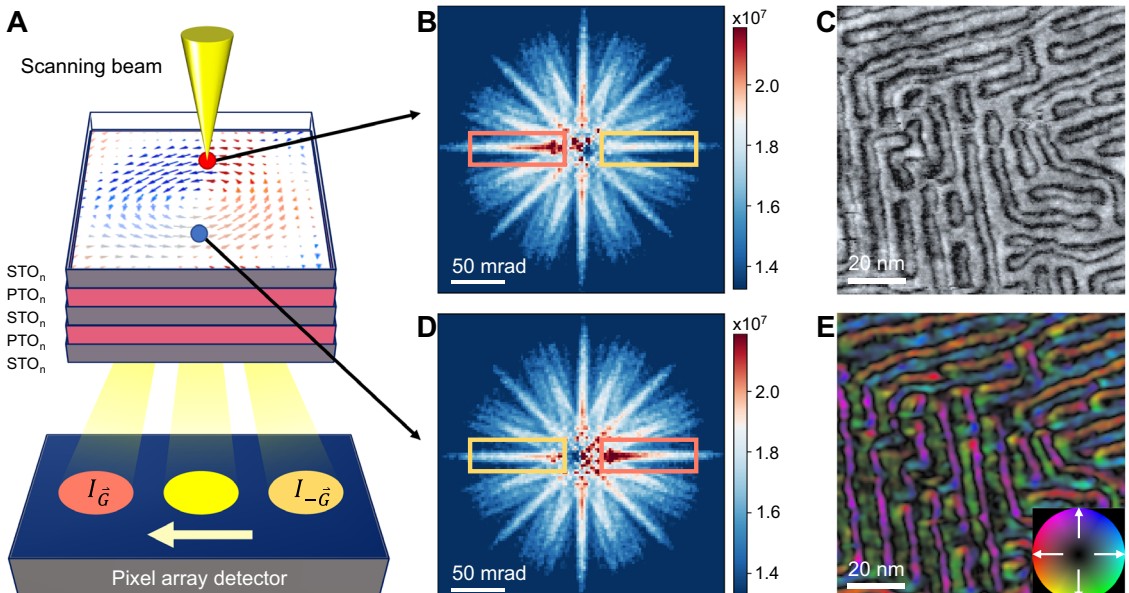

**Fig. 2 | Imaging in-plane polarization textures. A** Schematic of the plan-view 4D-STEM imaging technique on the $[(PbTiO_3)_{16}/(SrTiO_3)_{16}]_8$ superlattice which uses a scanning electron probe and pixelated array detector, where a diffraction pattern was recorded at each probe position. The local polarization direction can be determined by observing the difference of diffracted intensities of Friedel pairs, $I_{\vec{G}}$ and $I_{-\vec{G}}$. Representative diffraction patterns taken from (**B**) top and (**D**) bottom of a skyrmion, where the polarity-sensitive Kikuchi bands in the thermal diffuse scattering are selected for determining polarization, as marked by pink and yellow boxes. For clarity of display, the Kikuchi bands intensity were weighted by $k^2$, where k denotes the scattering vector from the transmitted spot. **C** Plan-view dark field STEM image of a $(SrTiO_3)_{16}/(PbTiO_3)_{16}/(SrTiO_3)_{16}$ trilayer reconstructed from the 4D-STEM dataset showing nanometer-size round and elongated features. **E** Polarization map reconstructed using Kikuchi bands from the same region as **C** showing the in-plane Bloch components of polar-skyrmions. The color map represents the in-plane polarization direction at each point.

STEM works by using an EMPAD detector which records the 2D electron diffraction pattern over a 2D grid of probe positions, resulting in 4D datasets (Fig. 2A, details in Methods)[38–40]. As a result of dynamical diffraction effects, the charge redistribution associated with ferroelectric polarization leads to the breakdown of Friedel's law[41,42]. Thus, due to channeling effects (Fig. S2), the polarization field within the top PbTiO₃ layer can be measured quantitatively from intensity differences of polarity-sensitive Kikuchi bands[43,44] (Fig. 2B, D) or Bragg reflections[45,46] (Fig. S3). We note that due to different channeling conditions, for plan-view imaging, atomic-resolution STEM is more sensitive to the top of the skyrmion, where the Néel component dominates, while the 4D-STEM method is more sensitive to the middle of the skyrmion where Bloch component dominates. To illustrate these channeling effects, we have performed multislice simulations of HAADF-STEM (semi-convergence angle of 21.4 mrad) and 4D-STEM datasets (semi-convergence angle of 2.45 mrad) on a model polar skyrmion obtained from 2$^{nd}$-principles calculations (Fig. S4).We employed these Kikuchi bands for polarity mapping as they are less sensitive to artifacts such as disinclination strain or crystal mis-tilts, which are inevitable in ferroic oxides[47,48] (Figs. S5 and S6). These Kikuchi bands are generally weak features require detectors of high dynamic range which is only available until very recently[37]. The fast detectors also allow for the mapping of polar textures which outruns the charging effects caused by the focused electron beam. For example, Fig. 2C shows a high-angle annular dark-field (HAADF) image of the skyrmions from plan-view reconstructed from the 4D-STEM dataset, where the out-of-plane polarization ($P_{op}$) is separated by domain walls with circular or elongated features. At the same region (same 4D-STEM dataset), Fig. 2E shows the in-plane polarization ($P_{ip}$) map of Bloch-like rotation can be reconstructed by using the polarity-sensitive Kikuchi bands. The white arrows and colors in Fig. 2E denote the direction of $P_{ip}$, whereas the saturation represents the vector magnitude. The dark color indicates the $P_{op}$ regions.

## Observation of topological phase transition in polar textures

Upon heating from 223 K to 373 K, we observed successive structural transitions in the skyrmion ensemble, from striped to circular-shaped polar skyrmions to a tetratic-ordered lattice. Figure 3A shows the polarization configuration at 223 K, which consists of elongated stripes of ~100 nm in length. In analogy with in-plane cuts ($Q_x$ – $Q_y$) from X-ray reciprocal space maps (RSM), the fast Fourier transform (FFT) patterns of plan-view HAADF images indicate the in-plane ordering of polar textures (Fig. S7). For example, two peaks were found in the FFT pattern along the pseudocubic $\pm(100)_{pc}$ directions (inset, Fig. 3A) consistent with the stripe features.

When heated to 298 K, the stripes deformed into circular shapes (Fig. 3B, E) of ~10 nm in diameter. The packing of circular skyrmions appears to be random and isotropic, i.e., no preferred orientational order, as indicated by the halo in the FFT pattern (inset, Fig. 3B). With further increase of temperature to 373 K, the random skyrmion arrangements were replaced by an ordered tetratic arrangement (Fig. 3C), confirmed by the four peaks in $\pm(100)_{pc}$ and $\pm(010)_{pc}$ directions in the corresponding FFT. Figure 3D-F show the corresponding polarization maps at different temperatures obtained from 4D-STEM datasets. For the stripe- (223 K) and disordered, circular-shaped (298 K) skyrmions, the polarization maps (Fig. 3D, E) indicate that the maximum $P_{ip}$ is observed at the periphery of the skyrmions, whereas the minima (almost zero) are at the core and outside the boundaries. Figure 3G−I show the corresponding phase-field simulations of polar textures at different temperatures and under various strain conditions, in which the average effective in-plane lattice constants are obtained from experimental measurements of >80,000 convergent beam electron diffraction (CBED) patterns (Fig. S8). The phase field simulations demonstrate a systematic change from the labyrinthine skyrmions at low temperature (223 K) to an ordered tetratic structure at higher temperatures (373 K), driven not by a pure thermal effect but by the changes in the in-plane strain state originating from lattice thermal expansion.

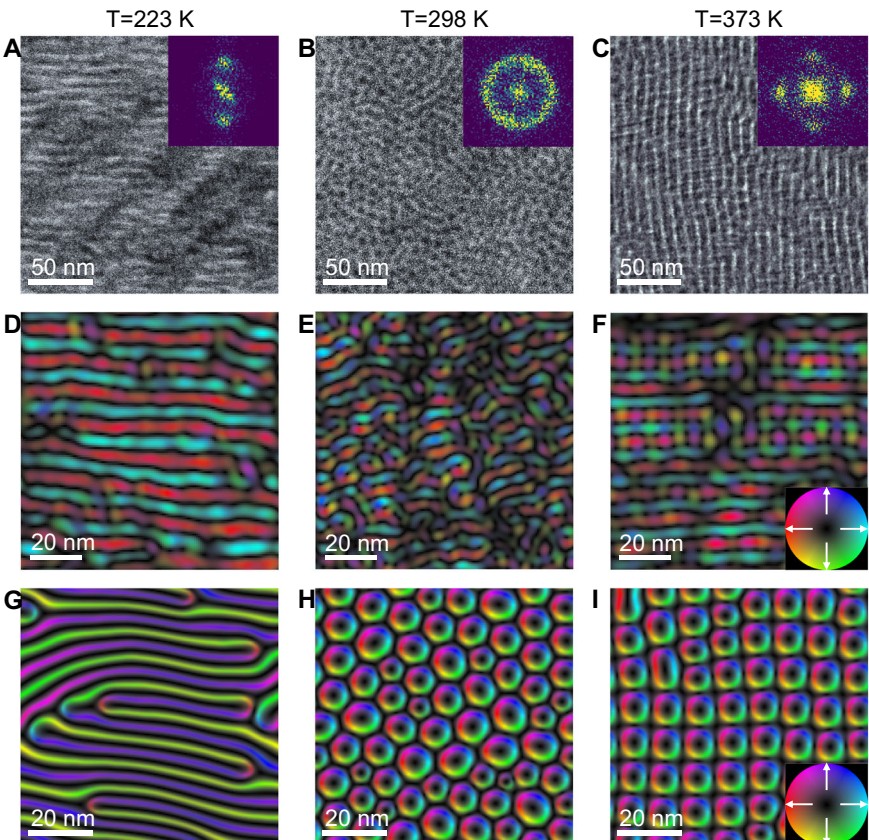

**Fig. 3 | Variations of polar textures with temperature.** Plan-view dark field STEM imaging of [(SrTiO₃)₁₆/(PbTiO₃)₁₆]₈ superlattice acquired with temperatures at (**A**) 223 K, (**B**) 298 K, and (**C**) 373 K. Insets, fast Fourier transform (FFT) of the images in **A**–**C** showing different types of ordering. Polarization maps reconstructed from the 4D-STEM dataset of superlattice at (**D**) 223 K, (**E**) 298 K, and (**F**) 373 K, showing the in-plane Bloch components of polarskyrmions. **D**–**F** are acquired from nearby regions of **A**–**C**. The corresponding phase-field simulations with temperatures and in-plane lattice parameters

(a,b) of (**G**) 223 K, $a = 3.875$ Å, $b = 3.885$ Å; **H** 298 K, $a = b = 3.905$ Å; and (**I**) 373 K, $a = b = 3.899$ Å. The color wheel hue (saturation) corresponds to the direction (magnitude) of the in-plane component of the ferroelectric polarization. We note that the phase field simulations capture the qualitative changes in morphology of particle-like objects as a function of temperature/strain conditions. However, while the general symmetry of the phase is predicted, the microscopic details of polar vectors at the boundaries require first-principles based calculations.

To compare the difference of polar textures, regions in Fig. 4A, B (yellow box) were selected to show the $P_{ip}$ polarization maps (Fig. 4C, D). The white arrows and colors in Fig. 4C, D denote the magnitude and the direction of $P_{ip}$, whereas the saturation represents the vector magnitude. The dark color indicates the $P_{op}$ regions at skyrmion cores and outside the boundary, which are separated by Bloch domain walls ($P_{ip}$) consistent with cross-section data (Fig. S9). The $P_{op}$ at the skyrmion cores points positively towards the growth direction ([001], or +z) and are antiparallel to $P_{op}$ outside of the boundary, labeled as green dots (+z) and red crosses (−z), respectively (Fig. 4E). The Bloch components ($P_{ip}$) exhibit a continuous rotation of the local polarization vector forming a closed loop, as illustrated in a map of the curl of the polarization vector field ($\nabla \times \mathbf{P}$)$_{[001]}$ (Fig. 4A). Both elongated and circular skyrmions exhibit $P_{ip}$ having clockwise (CW) rotation at the periphery yielding a vorticity of +1. Combining $P_{ip}$ and $P_{op}$ information, we confirm that both elongated and circular skyrmions manifest with a skyrmion number of $N_{sk} = −1$.

The most striking observation is the appearance of an ordered structure at 373 K (Fig. 4D, F), which represents a square meron lattice with $N_{sk} = −1/2$. Figure 4F clearly demonstrates these periodic arrays, which also indicates that the maximum $P_{ip}$ polarization is observed at the periphery, whereas the minimum (almost zero) is at the core. Three types of core regions were observed, which we labeled according to the cores of $P_{ip}$ having CW rotation (green), counterclockwise rotation (CCW; blue), and antivortices (red). The

dot in the circle indicates the $P_{op}$ pointed out of the page (along the growth direction), while the cross indicates P$_{op}$ pointing into the page. From a cross-section polarization map (Fig. S9), the $P_{op}$ at the cores of vortices (vorticity of +1) and antivortices (vorticity of −1) are plausibly antiparallel. On the other hand, $P_{op}$ appears at vanishing points of $P_{ip}$, implying the $P_{op}$ at vortex cores are not fully surrounded by $P_{op}$ of the opposite direction (Fig. S10). From this, we can compute the skyrmion number[3]:

$$N_{sk} = \frac{1}{4\pi} \iint d^2 r \vec{n} \cdot \left( \frac{\partial \vec{n}}{\partial x} \times \frac{\partial \vec{n}}{\partial y} \right) = \frac{1}{2} v \cdot \left( P_{op}^p - P_{op}^c \right), \quad (2)$$

where $v$ represents the vorticity, $P_{op}^c$ the value of the out of plane polarization at the core and $P_{op}^p$ the value of the out of plane polarization at the periphery. We deduce the vortices (blue and green dots) to be merons with a topological number of $N_{Sk} = \frac{1}{2} \cdot 1 \cdot (0 - 1) = -\frac{1}{2}$, and the antivortices as merons[25] with $N_{Sk} = \frac{1}{2}(-1) \cdot (0 - (-1)) = -\frac{1}{2}$ (Fig. 1).

## Imaging chirality of individual polar texture

While 4D-STEM works well for mapping $P_{ip}$ polarizations, we have so far only speculated about the corresponding $P_{op}$ at skyrmion cores based on the cross-section data (Fig. S9). To test our hypothesis, it is necessary to experimentally determine the chirality of the 3D polarvector field for each skyrmion, which poses challenges for projection techniques such as TEM. Fortunately, we can overcome this problem

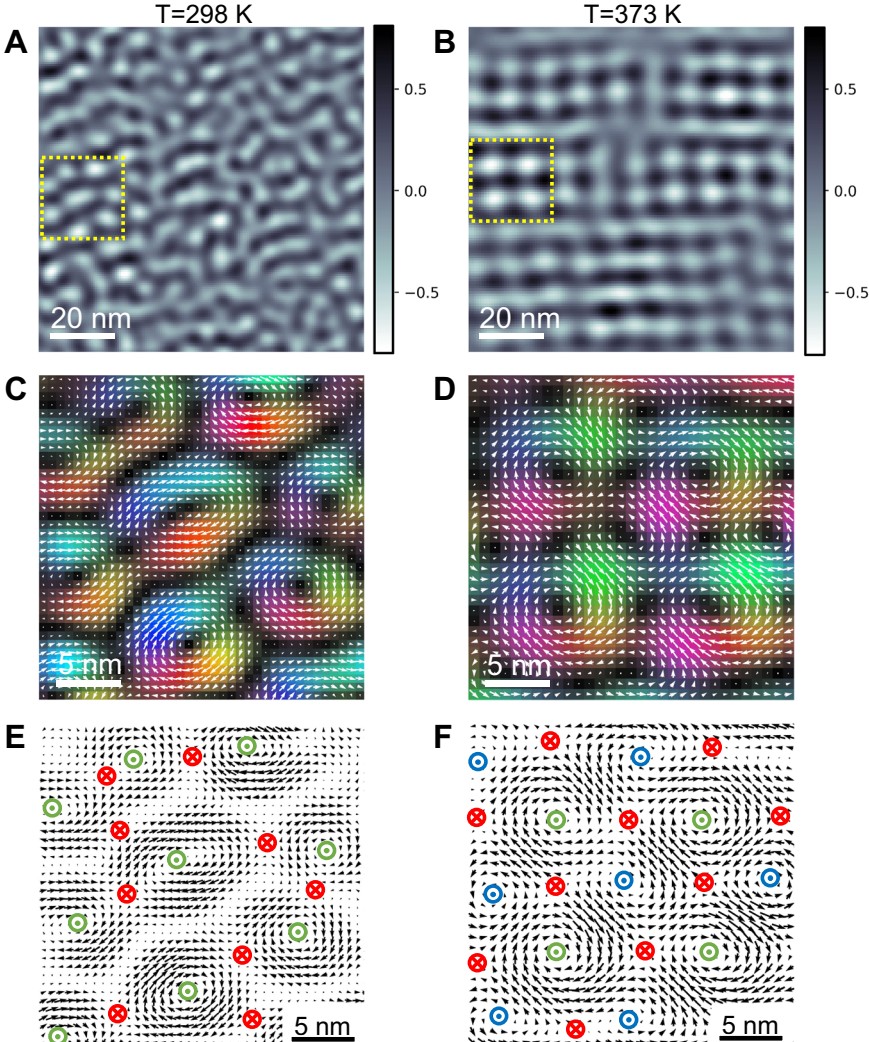

**Fig. 4 | Real-space observations of disordered polar skyrmions and a square lattice of merons.** The curl of in-plane polarization $(\bar{\nabla} \times P)_{[001]}$ showing the rotation directions of the structures at temperatures of (**A**) 298 K and (**B**) 373 K. **C**, **D** Enlarged in-plane polarization mapping from the yellow box region of A and B, respectively exhibiting the skyrmion texture at 298 K and local ordered meron textures at 373 K. **E**, **F** Details of (**C**, **D**), where vortices (clockwise: green, counterclockwise: blue) and antivortices (red) are labeled. The dots in circles represent polarization pointing out of the page, while the cross points into the page.

by utilizing the dynamical diffraction effects in higher-order Laue zone (HOLZ) reflections, which was established to retrieve 3D structural information such as handedness of chiral crystals[49,50]. In this study, we specifically examine the intensity differences of chirality sensitive Bijvoet pairs[51], such as $(671)/(\bar{6}\bar{7}\bar{1})$ and $(771)/(\bar{7}\bar{7}\bar{1})$ (yellow box, Fig. 5A).

First, we use dynamical diffraction simulation of a right-handed skyrmion and meron as a reference. For a right-handed skyrmion or meron, the diffracted intensity for $(\bar{6}71)$ is stronger than that of $(671)$ at this exact incident beam direction, while other pairs of reflections remain approximately symmetrical (Fig. 5C, black curve). With this in mind, we can determine the chirality of an individual skyrmion by comparing intensity variations of Bijvoet pairs in a 4D-STEM dataset. At 298 K, we carefully selected regions with minimal crystal mis-tilts and determined that both elongated and circular skyrmions are left-handed (Fig. S11). An ordered tetratic lattice appears upon heating to 373 K, as shown in the $(771)/(\bar{7}\bar{7}\bar{1})$ intensity ratio map (Fig. 5B). In this region, two types of chiral structures are identified, labeled as #1 (red) and #2 (blue). HOLZ intensity profiles show that $(\bar{6}71)$ is weaker than $(671)$ from the diffraction pattern averaged over meron type #1, and the opposite case for meron type #2 (Fig. 5C, red and blue curves, respectively). Using dynamical diffraction simulations as reference

(Fig. 5C, dashed gray curve), we can thus determine the chirality for meron type #1 as left-handed, and type #2 as right-handed. The alternating chirality of types #1 & #2 shown in Fig. 5B is consistent with our previous hypothesis of out-of-plane polarizations shown in Fig. 4F, labeled as green and blue vortices, respectively. More interestingly, the diffraction pattern averaged over a larger field of view (Fig. 5B, black box) shows symmetrical intensity, indicating the loss of chirality (Fig. 5C, black curve). Note that our observation does not violate the Poincaré-Hopf theorem[16,52], where a periodic array of vortices and antivortices yields zero net vorticity, yet imposing no constraints on the chiralities in such topological textures.

## Discussion

The transition from skyrmions to merons has been reported in chiral magnets with increasing in-plane magnetic anisotropy[2,30,53]. In 2D ferroelectrics, from the mesoscopic symmetry-breaking perspective, the observed isotropic-to-anisotropic structural transition upon heating was recently reported as an "inverse transition" of labyrinthine domains[22]. Such a transition was driven by the increase of thermal fluctuations of dipoles, which led to the domain reorientation or coalescence associated with the annihilation of merons and antimerons. In case of polar vortices, thermal fluctuations can lead to the

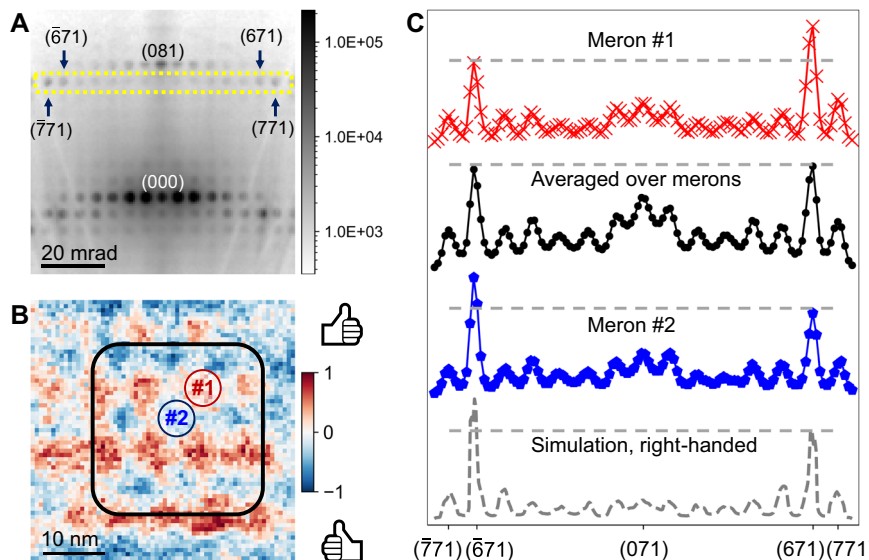

**Fig. 5 | Measurement of handedness of the chiral polar textures using 4D-STEM.**
**A** A representative experimental diffraction pattern acquired at 373 K (from the meron square lattice phase), oriented ~6.2° away from [001] zone axis, and tilted along one of the mirror planes. **B** Map of normalized intensity difference between (771) and ($\bar{7}$71) reflections reconstructed from the 4D-STEM dataset—i.e. one diffraction pattern is recorded at each spatial pixel in **B**. The positive (negative) regions indicate the polar textures having left-handed (right-handed) chirality. **C** Intensity line profiles of HOLZ reflections from diffraction patterns taken in from the selected regions #1, #2 labeled in **B**, displaying the intensity difference between two pairs of reflections: (671) and ($\bar{6}$71), (771) and ($\bar{7}$71). The colored labels in **B** indicates the region from which the diffraction patterns of **C** were extracted. The colored diffraction intensity profiles with markers show experimental data, whereas the simulation of a right-handed meron is shown in dashed gray line at the bottom. The dashed yellow box in **A** shows the region of diffraction space integrated to produce the HOLZ line profiles in **C**.

melting of chiral crystals and even the loss of chirality[54]. On the other hand, in the case of polar skyrmions, the origin of the observed transition from skyrmions to a meron lattice can be attributed to changes in the elastic boundary conditions. The strain imposed on the [(PbTiO₃)₁₆/(SrTiO₃)₁₆]₈ lifted-off membranes is caused by the thermal stress upon heating/cooling. In our experiment, the oxide membranes are ~4× higher in thermal expansion coefficients than the SiNₓ TEM grid to which they are attached[55,56]. The oxide membranes are suspended over a 2 μm hole, the strain is applied from the edge of the hole due to thermal stress, and therefore small locally varying anisotropy within which gives small local bias to the texture orientation. As a consequence, the lifted-off membrane is under compressive strain at 373 K and tensile strain at 223 K, with additional local bending. In case of 4D-STEM experiments, we deliberately searched for flat regions to avoid artifacts in measuring $P_{ip}$. A detailed 4D-STEM analysis of averaged in-plane lattice parameters (>80,000 diffraction patterns) shows that the lifted-off membrane is locally more rectangular at 223 K than at 373 K (Fig. S8). Experimentally, the lattice constant is strongly temperature dependent. To decouple the influence of strain versus purely thermal effects on the phase transition, phase-field simulations were carried out at controlled strain boundary conditions for the range of measured temperatures (Fig. S12). For a fixed in-plane lattice constant, simulation results for temperatures from 223–373 K showed a similar disordered skyrmion phase, indicating that here strain plays an important role in the skyrmion ordering. Furthermore, the change in anisotropy of in-plane lattice parameters, as seen in the 4D-STEM strain analysis (inset, Fig. S8D), is also consistent with FFT patterns for stripes and square lattice (Fig. 3A and C). Thus, we anticipate the occurrence of a long-range ordered meron lattice at room temperature in (PbTiO₃)ₙ/(SrTiO₃)ₙ superlattices grown on a compressive substrate such as LaAlO₃.

In summary, we report the observation of a topological phase transition sequence in polar skyrmions, whereby tuning the temperature and strain boundary conditions, the anisotropic stripe phase deforms into an isotropic disordered circular phase, and finally transforms into an anisotropic ordered phase. This is the first observation of such transitions in a dipolar topological structure, in which the skyrmions deform in shape and a square lattice of merons appears, thereby preserving the topological charge of the system. The chiralities for each phase were also determined experimentally at nm-scale, for the first time. We hope that the microscopic observation of such a topological phase transition will stimulate further work to explore the macroscopic manifestation of the changes in topology with strain, which clearly is the critical external stimulus (in contrast to magnetic systems, where the Dzyalozhinskii-Moriya coupling plays a key role). Finally, our findings imply that such dipolar textures are a fertile ground for exploring new phases and topology, and with possible applications in nanoscale ferroelectric logic and storage devices.

## Methods

### Deposition of thin film superlattices and membrane lift-off

The epitaxial lift off sacrificial layer of 16 nm Sr₂CaAl₂O₆ with a 2.4 nm SrTiO₃ capping layer was synthesized on single-crystalline SrTiO₃ (001) substrates via reflection high-energy electron diffraction (RHEED) – assisted pulsed laser deposition. Subsequent to this, *n*-SrTiO₃/*n*-PbTiO₃/*n*-SrTiO₃ trilayers (*n*- is the number of monolayers; *n* = 16) and [(PbTiO₃)₁₆/(SrTiO₃)₁₆]₈ superlattices were synthesized ex-situ on this template via RHEED-assisted pulsed-laser deposition (KrF laser). RHEED was used during the deposition to ensure the maintenance of a layer-by-layer growth mode for both the PbTiO₃ and SrTiO₃. (*Sample Preparation using RHEED-assisted Pulsed-laser Deposition*). The specular RHEED spot was used to monitor the RHEED oscillations. The heterostructure was first spin-coated with a polymer support of 500 nm thick polymethyl methacrylate (PMMA) film and placed in deionized water at room temperature until the sacrificial Sr₂CaAl₂O₆ layer was fully dissolved. The PMMA coated film was then released from the substrate and transferred onto the TEM grid. Finally, the PMMA layer was dissolved and removed from the membrane in acetone. Additional details can be found in *Membrane Lift-off and Transfer*.

## 4D-STEM for Polarization and Chirality determination

We performed 4D-STEM experiments using an electron microscopy pixel array detector (EMPAD), where the 2D electron diffraction pattern was recorded over a 2D grid of real space probe positions, resulting in 4D datasets. Due to dynamical diffraction effects, the charge redistribution associated with polarization leads to the breakdown of Friedel's law (*4D-STEM for Polarization Mapping*). From the collected CBED patterns, the polarization direction in the plan-view samples is reconstructed by calculating the center-of-mass in polarity sensitive Kikuchi bands along the cubic directions. We employ the Kikuchi bands as a more robust means to extract polarity information against internal crystal mis-tilts, which often occurs in ferroic oxides due to disinclination strain. Since chirality is a 3D property and TEM is a projection technique, we can retrieve this 3D information by utilizing the intensity asymmetries of chirality sensitive Bijvoet pairs in higher order Laue zone (HOLZ) reflections. Additional details for chirality determination are discussed in *4D-STEM for Chirality Determination*.

## Phase-field simulations

Detailed expressions of the energy terms, materials parameters as well as the numerical simulation procedure is described in ref(10). A periodic boundary condition is used along the in-plane dimensions, while a superposition method is applied in the out-of-plane direction. To determine the local strain state, the reference pseudocubic lattice constants for STO (PTO) at 223 K, 300 K and 373 K are set as 3.901 Å (3.953 Å), 3.905 Å (3.957 Å) and 3.909 Å (3.961 Å), respectively. The average effective substrate lattice constants are taken from experimental measurements, which are set as $a = 3.875$ Å, $b = 3.885$ Å for 223 K, $a = b = 3.905$ Å for 300 K and $a = b = 3.899$ Å for 373 K. The large compressive strains are due to local bending when the lift-off membrane is heated/cooled down. We believe that the dominant effect is the global thermal induced large compressive strain under heating, thus, for the simplicity of the model, we ignore the local strain inhomogeneity. A background dielectric constant of 40 is used. Random noise with a magnitude of 0.0001 C/m$^2$ is added to the system as the initial polarization distribution of the system. Additional details of the simulation parameters and the resulting structures are discussed in *Phase-Field Calculations*.

## Data availability

Experimental and simulation data (4D-STEM dataset and python script for analysis) are provided in the figures and Supplementary Information, and are publicly available at https://doi.org/10.5281/zenodo.7500030. Additional data are available from the corresponding authors upon request.

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

## Acknowledgements

The authors acknowledge fruitful discussions with Prof. Jian-Min Zuo, Prof. Kin Fai Mak, Dr. Shengwei Jiang, and Zui Tao. Funding was primarily provided by the Department of Defense, U.S. Army Research Office under the MURI ETHOS, via cooperative agreement W911NF-21-2-0162 (R.R., D.M., Y.-T.S.); and the U.S. Air Force Office of Scientific Research Hybrid Materials MURI, under award no. FA9550-18-1-0480 (H.Y.H.). The electron microscopy studies were performed at the Cornell Center for Materials Research, a National Science Foundation (NSF) Materials Research Science and Engineering Center (DMR-1719875). The Cornell FEI Titan Themis 300 was acquired through NSF-MRI-1429155, with additional support from Cornell University, the Weill Institute and the Kavli Institute at Cornell. The authors thank M. Thomas, J. G. Grazul, M. Silvestry Ramos, K. Spoth for technical support and careful maintenance of the instruments. The materials synthesis work is supported by the Quantum Materials program from the Office of Basic Energy Sciences, US Department of Energy (DE-AC02-05CH11231). The membrane lift-off techniques were developed with support from US Department of Energy, Office of Basic Energy Sciences, Division of Materials Sciences and Engineering, under contract number DE-AC02-76SF00515. The phase-field simulation work is supported as part of the Computational Materials Sciences Program funded by the U.S. Department of Energy, Office of Science, Basic Energy Sciences, under Award No. DE-SC0020145. F.G.-O., P.G.-F., and J.J. acknowledge financial support from Grant No. PGC2018-096955-B-C41 funded by MCIN/AEI/10.13039/501100011033 and by ERDF "A way of making Europe," by the European Union. F.G.-O. acknowledges financial support from Grant No. FPU18/04661 funded by MCIN/AEI/10.13039/501100011033. S.D. acknowledges Science and Engineering Research Board (SRG/2022/000058) and Indian Institute of Science start up grant for financial support. L.W.M. acknowledges sup-port from the U.S. Department of Energy, Office of Science, Office of Basic Energy Sciences, under Award Number DE-SC-0012375 for the development and study of ferroic heterostructures.

## Author contributions

Y.-T.S., S.D., R.X., H.Y.H., R.R., and D.M. designed research; Y.-T.S., S.D., Z.H., R.X., S.C., F.G.-O., and P.G.-F., and J.J. performed research; Y.-T.S., S.D., and R.X. contributed new reagents/analytic tools; Y.-T.S. super-vised by D.M., S.D., Z.H., F.G.-O., P.G.-F., L.-Q.C., H.Y.H., J.J., L.W.M., and R.R. analyzed data; and Y.-T.S and D.M. wrote the paper.

## Competing interests

The authors declare no competing interests.
