## [Peer Review File · Nature Communications]

Emergent chirality in a polar meron to skyrmion phase transitionReviewers' Comments:

Reviewer #1:

Remarks to the Author:

In this manuscript, Shao et al reported the strain (temperature)-driven phase transition from the stripe-like domains to skyrmion bubbles, to a meron lattice in the lifted-off [(PbTiO₃)₁₆/(SrTiO₃)₁₆]₈ superlattice membranes. This is the first observation of such transitions in this system. The results are very novel and could further stimulate the exploration of emergent ferroelectric topological structures and their manipulation. The 4DSTEM technique used for the determination of chirality and domain structures is also very efficient and robust against the defects and mis-tilting, in contrast to the conventional real-space atomic position analysis. Therefore, I recommend its publication after the following questions are addressed.

1. The authors said that both in-plane and out-plane polar order can be mapped simultaneously. However, I noticed that only the projected in-plane polarization can be directly measured by analyzing the 4DSTEM Kikuchi bands. Also, no 3D structure of these topological objects is given in the manuscript.
2. The dipolar textures in Figure 1 in not very clear for the readers, especially Figure 1E. There are so many beautiful sketches for the topologically magnetic objects. I suggested the authors refer to these magnetic objects and plot the dipolar merons with more arrows to guide the eyes.
3. The authors attributed the thermal induced strain to the observed phase transitions. Although it is well reproduced by the phase field simulations, the pure thermal effects on the polarization are not discussed, as explained in the recent papers [arXiv:2203.03522, 2022.]. How did the authors exclude this?
4. The polarization mapping is measured by the developed 4DSTEM based on the Kikuchi bands. As atom position analysis is the most popular and direct way based on the high-resolution STEM images, the authors should also present the consistent results by the latter method for stripe-like, circular-shaped and squared shaped structures. Atomic scale observations of these structures are necessary.
5. Will the diffraction contrast affect the intensity distribution of Kikuchi bands? The robustness of Kikuchi bands based 4DSTEM techniques should be discussed and simulated in details.

Reviewer #2:

Remarks to the Author:

In this manuscript, the authors reported the observation of phase transition and the chirality in a ferroelectric superlattice membranes using a combination of experimental and theoretical works. SCBED was introduced to map the local polarization, strain and the chirality of the skyrmions. Elastic boundary conditions were imposed by varying temperatures in free-standing membrane, and the structural transition of chiral polar skyrmions to non-chiral merons was demonstrated. The phenomenon observed in this manuscript is instructive for the community. I recommend publish this work in Nature Communications after the following comments were addressed by the authors:

1. The authors claimed that the polarization field within the top PTO layer can be measured using SCBED due to channeling effects, which is supported by Fig.S2 (the same data with the previous Nature paper, reference No.19 in the present manuscript) and some related references. However, the kinetical scattering is complicated in such a thick membranes (>100nm), the reliability of SCBED data analysis should be enhanced by thinner sample comparisons, such as three-layered ones, and more imaging simulations should be added.
2. At line 1 in page 5, "Although phase field models had predicted possibility of the forming a long-range ordered skyrmion lattice for certain values of mismatch strain with the substrate¹⁹, ..." However, this result is not shown in the reference 19 (Das, S. et al. Nature 2019).
3. At line 22 in page 5, the authors claimed "the first time...", however, the chirality of individual polar skyrmion was already demonstrated in the previous Nature paper (Das, S. et al. Nature 2019).
4. In page 8 and 13, the compressive or tensile strain were introduced due to the thermal expansion difference between the grid and the suspended membrane. This view is mostly speculation. Other

factors should be excluded, such as grid bending, phase transition, etc. It is recommended to supplement more temperature effect experiments.

5. The authors use two kinds of lifted-off free-standing films, $(\text{SrTiO}_3)_{16}/(\text{PbTiO}_3)_{16}/(\text{SrTiO}_3)_{16}$ trilayer and $[(\text{PbTiO}_3)_{16}/(\text{SrTiO}_3)_{16}]_8$ superlattice membranes, for comparison? Please make further explain.

6. In the following sentences, "we demonstrate strain as a crucial order parameter to drive isotropic-to-anisotropic structural transitions of chiral polar skyrmions to non-chiral merons, ..." , " we report the observation of a topological phase transition sequence in polar skyrmions, whereby tuning the temperature and strain boundary conditions, the anisotropic stripe phase deforms into an isotropic disordered circular phase, and finally transforms into an anisotropic ordered phase." How to understand anisotropic, isotropic and isotropic-to-anisotropic structural transitions? Please clarify more about the way of stress loading or introduction in the process of changing temperature.

7. The 4D-STEM method for describing the skyrmion polarization has been used in the previous Nature article (Das, S. et al. Nature 2019). The Figure 1 in the Nature paper shows that the three-layer sample is a striped structure, and the superlattice sample shows random skyrmion domains. However, the FFT image in Fig. 3C in the present manuscript is similar to RSM image in Figure 1 and Extended Data Fig.1 in the Nature paper. Please clarify them, and what is the main reason for the transition from strip-shaped elongated features to circular-shaped circular features, the temperature or the number of layers? Figure S1 also shows that one is a long strip and the other is a circle. What is the mechanism of this different shapes? It is recommended to supplement the temperature-variable experiment in the three-layer structure.

8. Finally, in the Abstract, the description "This topological phase transition is accompanied by a change in chirality, from net-handedness (in skyrmionic phase) to zero-net chirality (in meronic phase)." looks a little inconsistent with the title "Emergent chirality in a polar meron to skyrmion phase transition".

Reviewer #3:

Remarks to the Author:

In this paper, the authors employed state-of-the-art experiment and simulation approaches to investigate the nontrivial topological polar textures and the associated phase transition in the oxide superlattice thin films. In the lift-off PTO/STO superlattices, they employed 4D-STEM to detect skyrmions and melons structures and developed new analysis method to determine the chiralities of each phases in experiments for the first time. In addition, they also reported the observation of topological phase transition sequences, whereby tuning the temperature and strain boundary conditions (induced by thermal expansion mismatch) conditions, the anisotropic stripe phase transformed to isotropic disordered circular phase, and finally transformed into anisotropic (square meron lattice). This is also the first experimental observation. Although some interesting questions were raised up instead of being answered in this work, it will certainly stimulate further research works to explore the macroscopic manipulation of the topological textures. This referee recommends the acceptance after the authors address one minor comment.

The phase field simulation plays an important role in this manuscript to understand the transition (e.g., strain driven) observed in experiments. But the details are missing. What is the geometry of the simulation model? Is it bilayer materials, substrate bottom and oxide superlattice on top? In experiments, the thin film is placed over a pin-hole. The strain field could be non-uniform. It could be complicated if the z-direction relaxation was considered. Did the authors consider this in their modelling? Did they believe this detail is not significant for this study? Some discussion should be provided.

Response Letter for Emergent Chirality in a Polar Meron to Skyrmion Transition

We appreciate the reviewers' support. We also are grateful for the constructive feedback and efforts to strengthen the manuscript. Specifically, we have addressed the following:

- To address Reviewer #1's questions, we have clarified that the role of thermal fluctuations in the skyrmions and compared with the case of vortices.
- To address both Reviewers #1 & #2's questions, we have compared the difference in atomic-resolution STEM and 4D-STEM for polarization imaging performing new, extensive multislice simulations of a model polar skyrmion to provide a ground truth for both methods.
- To address Reviewer #3's question, we have included the phase-field simulation details and considered geometries for elucidating strain versus temperature effects.

Our point-by-point responses to the reviewers' comments are listed below, with our reply written in blue and excerpts from the revised text in green. Edits and additions to the Main Text and Supplementary Information have been highlighted in yellow.

Reviewer #1 (Remarks to the Author):

In this manuscript, Shao et al reported the strain (temperature)-driven phase transition from the stripe-like domains to skyrmion bubbles, to a meron lattice in the lifted-off [(PbTiO₃)₁₆/(SrTiO₃)₁₆]₈ superlattice membranes. This is the first observation of such transitions in this system. The results are very novel and could further stimulate the exploration of emergent ferroelectric topological structures and their manipulation. The 4DSTEM technique used for the determination of chirality and domain structures is also very efficient and robust against the defects and mis-tilting, in contrast to the conventional real-space atomic position analysis. Therefore, I recommend its publication after the following questions are addressed.

1. The authors said that both in-plane and out-plane polar order can be mapped simultaneously. However, I noticed that only the projected in-plane polarization can be directly measured by analyzing the 4DSTEM Kikuchi bands.

It is correct that the 4D-STEM Kikuchi bands method measures only the projected in-plane component of polarization. However, the out-of-plane information needed to determine chirality—a three-dimensional property—was obtained by analysis of the high-order Laue zone (HOLZ) reflections, also from the 4D-STEM datasets, in which the HOLZ carries the out-of-plane information with finite (k_x , k_y , k_z) components. This was shown in figure 5, with Fig 5b showing the measured real-space map of 3D chirality, which requires simultaneous in-plane and out-of-plane information, provided by the chosen HOLZ reflections. Out of plane polarization at the atomic scale was given in figure S9.

Also, no 3D structure of these topological objects is given in the manuscript.

We thank the reviewer for the suggestion. To clarify, we did include the schematic 3D structures of polar skyrmions and merons in Fig. 1 of the current manuscript. We have modified Fig. 1 for better clarity, adding an anti-skyrmion as figure 1c to show the effect of the out-of-plane component. When viewed in projection, the RH skyrmion (B) and the LH anti-skyrmion (C) are indistinguishable due to the same winding in-plane components – they differ only in their out-of-plane components.

2. The dipolar textures in Figure 1 in not very clear for the readers, especially Figure 1E. There are so many beautiful sketches for the topologically magnetic objects. I suggested the authors refer to these magnetic objects and plot the dipolar merons with more arrows to guide the eyes.

We thank the reviewer for the suggestions. As noted above, we have added detail to the sketches in Figure 1, with finer arrows and details in each of the polar textures for better clarity. We note that the purpose of the illustrations is to show the differences between a skyrmion and a meron, which lie in the evolution of the out-of-plane components from core to periphery, and we have added additional detail to figures 1E-F to highlight this.

3. The authors attributed the thermal induced strain to the observed phase transitions. Although it is well reproduced by the phase field simulations, the pure thermal effects on the polarization are not discussed, as explained in the recent papers [arXiv:2203.03522, 2022.]. How did the authors exclude this?

It is important to note that the systems discussed in the paper quoted by the reviewer (1-dimensional polar vortices) and the one discussed in this manuscript (0-dimensional polar skyrmions) are different. We expect a difference in phase diagram due to the different mechanical boundary conditions (i.e., epitaxial strain) and the differences in dimensionality (0D vs 1D).

In this work, the purpose of the phase field simulations was to understand and decouple pure strain from pure temperature effects that cannot be done experimentally since the lattice constant is strongly temperature dependent. We find that the effect of temperature (decoupled from strain) within the temperature range of 223-373 K is negligible as demonstrated by the phase-field simulations, in Figs. 3G-I and Fig. S12, and the corresponding text in page 13, lines 16-20 in the Main Text:

“Experimentally, the lattice constant is strongly temperature dependent. To decouple the influence of strain versus purely thermal effects on the phase transition, phase-field simulations were carried out at controlled strain boundary conditions for the range of measured temperatures (Fig. S12). For a fixed in-plane lattice constant, simulation results for temperatures from 223-373K showed a similar disordered skyrmion phase, indicating that here strain plays an important role in the skyrmion ordering.”

4. The polarization mapping is measured by the developed 4DSTEM based on the Kikuchi bands. As atom position analysis is the most popular and direct way based on the high-resolution STEM images, the authors should also present the consistent results by the latter method for stripe-like, circular-shaped and squared shaped structures. Atomic scale observations of these structures are necessary.

High-resolution STEM is indeed a popular and direct way for imaging polarization in structures that do not vary in projection, however the atomic-resolution signal dechannels quickly and thus is sensitive only to atomic distortions near the entrance surface of the sample. To probe deeper into the sample, we developed the new 4D-STEM techniques by utilizing the different channeling condition due to a smaller probe semi-convergence angle.

To quantify the multiple scattering effects, we have added additional multislice simulations of HAADF-STEM and 4D-STEM datasets on a model polar skyrmion structure. The polar skyrmion structure in a $(\text{SrTiO}_3)_{16}/(\text{PbTiO}_3)_{16}/(\text{SrTiO}_3)_{16}$ trilayer structure (thickness ~ 19 nm) was obtained from 2nd-principles calculations. The extracted polar vectors from both HAADF-STEM and 4D-STEM were then compared qualitatively with the polar textures at different z-slices of the xy-planes in a 3D polar skyrmion. Indeed, the HAADF-STEM and 4D-STEM results qualitatively agree with the top (Néel components) and middle (Bloch components) slices of the skyrmion, respectively. We added this new Fig. S4 and the corresponding text in page 6, lines 15-20:

“We note that due to different channeling conditions, for plan-view imaging, atomic-resolution STEM is more sensitive to the top of the skyrmion, where the Néel component dominates, while the 4D-STEM method is more sensitive to the middle of the skyrmion where Bloch component dominates. To illustrate these channeling effects, we have performed multislice simulations of HAADF-STEM (semi-convergence angle of 21.4 mrad) and 4D-STEM datasets (semi-convergence angle of 2.45 mrad) on a model polar skyrmion obtained from 2nd-principles calculations (Fig. S4).”

Fig. S4. Plan-view polarization mapping of a polar skyrmion using atomic-resolution HAADF-STEM versus 4D-STEM Kikuchi bands. Simulated images for both (A) atomic resolution HAADF-STEM and (B) ADF-STEM image generated from the multislice-calculated 4D-STEM dataset. (C) Ti-displacement vector map obtained from (A), and (D) polarity map obtained from Kikuchi bands in the simulated 4D-

STEM dataset in (B). The in-plane polarization component obtained from second-principles calculations of the skyrmion structure showing (E) hedgehog-like structure at the top and (F) Bloch vortex from weighted projection with more on the central plane in PbTiO_3 . Due to difference in channeling conditions (the more convergent HAADF-STEM dechannels much closer to the entrance surface), HAADF-STEM and 4D-STEM are sensitive to Néel and Bloch components of the polar skyrmion, respectively.

5. Will the diffraction contrast affect the intensity distribution of Kikuchi bands? The robustness of Kikuchi bands based 4DSTEM techniques should be discussed and simulated in details.

We thank the reviewer for this clarifying question. In both HAADF-STEM and Bragg-beam-based 4D-STEM imaging, one major source of confounding artifacts is the crystal mis-tilts, which are inevitable as they are the consequence of lattice disclinations at the domain boundaries in ferroic oxides. The crystal mis-tilt will affect the diffraction condition and hence possibly induce contrast reversal in the polarity-sensitive Bragg reflections with tilts as small as 0.04° .

These contrast reversals can be greatly suppressed with the Kikuchi band method. For instance, small tilts from the zone axis simply causes a shift in the Kikuchi bands with the intensity asymmetry preserved up to 0.23° . To illustrate this, we simulated both polarity sensitive Bragg reflections and Kikuchi bands in PbTiO_3 as a function of mis-tilt. We added this new Fig. S6 and attached as below:

Fig. S6. Diffraction effects and mis-tilt artifacts on the polarity determination of PbTiO_3 . Due to dynamical diffraction effects, the polarity-sensitive (A) Bragg reflections (e.g., $\pm(001)$, $\pm(002)$, and $\pm(003)$) and (B) Kikuchi bands (e.g., $\pm(001)$) along the polar axis exhibit intensity asymmetry. However, this intensity asymmetry can be largely affected by sample mis-tilt artifacts. Here, we simulate the CBED patterns of PbTiO_3 along $[100]_{\text{pc}}$ zone-axis with systematic crystal mis-tilt angles. (C) The intensity asymmetry in Bragg reflections can have contrast reversal with mis-tilts as small as 0.04° , while the Kikuchi bands in (D) can tolerate mis-tilt angles up to 0.23° . The diffraction patterns were simulated using 300 keV electrons with probe semi-convergence angle of 2 mrad, and PbTiO_3 thickness of 101 nm as consistent with the superlattice membrane. The intensity asymmetry is normalized as $I_{\text{asymmetry}} = \frac{(I_{hkl} - I_{\bar{h}\bar{k}\bar{l}})}{(I_{hkl} + I_{\bar{h}\bar{k}\bar{l}})}$

Reviewer #2 (Remarks to the Author):

In this manuscript, the authors reported the observation of phase transition and the chirality in a ferroelectric superlattice membranes using a combination of experimental and theoretical works. SCBED was introduced to map the local polarization, strain and the chirality of the skyrmions. Elastic boundary conditions were imposed by varying temperatures in free-standing membrane, and the structural transition of chiral polar skyrmions to non-chiral merons was demonstrated. The phenomenon observed in this manuscript is instructive for the community. I recommend publish this work in Nature Communications after the following comments were addressed by the authors:

We thank the reviewer for the encouraging comments on the novelty both in characterization methods and the observed phenomena described in this manuscript.

1. The authors claimed that the polarization field within the top PTO layer can be measured using SCBED due to channeling effects, which is supported by Fig.S2 (the same data with the previous Nature paper, reference No.19 in the present manuscript) and some related references. However, the kinetical scattering is complicated in such a thick membranes (>100nm), the reliability of SCBED data analysis should be enhanced by thinner sample comparisons, such as three-layered ones, and more imaging simulations should be added.

As suggested by the reviewer, we have performed additional multislice simulations of HAADF-STEM and 4D-STEM datasets on a theoretically calculated polar skyrmion to quantify the performance of the SCBED method in the presence of strong scattering. A model polar skyrmion structure for a $(\text{SrTiO}_3)_{16}/(\text{PbTiO}_3)_{16}/(\text{SrTiO}_3)_{16}$ trilayer structure (thickness ~19 nm) was obtained from 2nd-principles calculations. The extracted polar vectors from both full-multiple-scattering calculations of HAADF-STEM and 4D-STEM were then compared with the polar textures at different z-slices of the xy-planes in a 3D polar skyrmion. Indeed, the HAADF-STEM and 4D-STEM results qualitatively agree with the top (Néel components) and middle (Bloch components) slices of the skyrmion, respectively. We added this new Fig. S4 and the corresponding text in page 6, lines 15-20 in the Main Text:

“We note that due to different channeling conditions, for plan-view imaging, atomic-resolution STEM is more sensitive to the top of the skyrmion, where the Néel component dominates, while the 4D-STEM method is more sensitive to the middle of the skyrmion where Bloch component dominates. To illustrate these channeling effects, we have performed multislice simulations of HAADF-STEM (semi-convergence angle of 21.4 mrad) and 4D-STEM datasets (semi-convergence angle of 2.45 mrad) on a model polar skyrmion obtained from 2nd-principles calculations (Fig. S4).”

Fig. S4. Plan-view polarization mapping of a polar skyrmion using atomic-resolution HAADF-STEM versus 4D-STEM Kikuchi bands. Simulated images for both (A) atomic resolution HAADF-STEM and (B) ADF-STEM image generated from the multislice-calculated 4D-STEM dataset. (C) Ti-displacement vector map obtained from (A), and (D) polarity map obtained from Kikuchi bands in the simulated 4D-STEM dataset in (B). The in-plane polarization component obtained from second-principles calculations of the skyrmion structure showing (E) hedgehog-like structure at the top and (F) Bloch vortex from weighted projection with more on the central plane in PbTiO_3 . The HAADF-STEM and 4D-STEM datasets are simulated using a 300 keV electron, with probe semi-convergence angles of 21.4 mrad and 2.45 mrad, respectively. Due to difference in channeling conditions (the more convergent HAADF-STEM dechannels

much closer to the entrance surface), HAADF-STEM and 4D-STEM are sensitive to Néel and Bloch components of the polar skyrmion, respectively

2. At line 1 in page 5, “Although phase field models had predicted possibility of the forming a long-range ordered skyrmion lattice for certain values of mismatch strain with the substrate¹⁹, ...” However, this result is not shown in the reference 19 (Das, S. et al. Nature 2019).

We thank the reviewer for pointing out this error. We have corrected this reference to one of co-author’s Ph.D. thesis: Zijian Hong, Phase-Field Simulations of Topological Structures and Topological Phase Transitions in Ferroelectric Oxide Heterostructures, The Penn State University (2017) ISBN 978-0-355-33061-8. (<https://etda.libraries.psu.edu/catalog/14257zxh121>)

3. At line 22 in page 5, the authors claimed “the first time....”, however, the chirality of individual polar skyrmion was already demonstrated in the previous Nature paper (Das, S. et al. Nature 2019).

We note that the chiral signal identified in our previous Nature paper (Das, S. et al. Nature 2019) using resonant x-ray circular dichroism (RXCD) was an ensemble average of ~100,000 of polar skyrmions (which are not all the same handedness), while in this manuscript we demonstrated the chirality determination for individual polar skyrmions using 4D-STEM and HOLZ reflections (Fig 5). The difference can simply be explained by the illumination area of the probe, which is >10s μm^2 for x-rays and <1 nm^2 for electrons, and the size of a skyrmion is about 10 nm.

4. In page 8 and 13, the compressive or tensile strain were introduced due to the thermal expansion difference between the grid and the suspended membrane. This view is mostly speculation. Other factors should be excluded, such as grid bending, phase transition, etc. It is recommended to supplement more temperature effect experiments.

We agree the reviewers that there may be possibilities of bending of the membranes. Experimentally, we deliberately avoided the heavily-bent regions to obtain better quality 4D-STEM datasets, because large crystal mis-tilts cause serious imaging artifacts. We then calculated the relative strain among those 4D-STEM datasets (>80,000 diffraction patterns) obtained from relatively flat regions (crystal mis-tilts on the order of ~0.1°) and their histogram was shown in Fig. S5. Thus, within the field-of-view of ~200 nm, we can safely exclude the effects of grid bending.

The purpose of the phase field simulations was to understand and decouple pure strain from pure temperature effects that cannot be done experimentally since the lattice constant is strongly temperature dependent. We find that the effect of temperature (decoupled from strain) within the temperature range of 223-373 K is negligible as demonstrated by the phase-field simulations, in Figs. 3G-I and Fig. S12, and the corresponding text in page 13, lines 16-20 in the Main Text:

“Experimentally, the lattice constant is strongly temperature dependent. To decouple the influence of strain versus purely thermal effects on the phase transition, phase-field simulations were carried out at controlled strain boundary conditions for the range of measured temperatures (Fig. S12). For a fixed in-plane lattice constant, simulation results for temperatures from 223-373K showed a similar disordered skyrmion phase, indicating that here strain plays an important role in the skyrmion ordering.”

5. The authors use two kinds of lifted-off free-standing films, (SrTiO₃)₁₆/(PbTiO₃)₁₆/(SrTiO₃)₁₆ trilayer and [(PbTiO₃)₁₆/(SrTiO₃)₁₆]₈ superlattice membranes, for comparison? Please make further explain.

The purpose of showing the lifted-off membranes for both trilayer and superlattice of 8 repeats is 1) to compare their polar textures, as well as 2) to illustrate the different types of diffraction information employed for polarization imaging in 4D-STEM datasets. First, as a confirmation of morphologies reported in the Nature paper (Das, S. et al. Nature 2019), the trilayer and superlattice exhibit labyrinth and skyrmion textures, respectively. Second, we note that due to the difference in sample thicknesses, the suitable 4D-STEM methods for plan-view polarization imaging of trilayer and superlattice membranes are polarity-sensitive Bragg reflections and Kikuchi bands, respectively.

6. In the following sentences, “we demonstrate strain as a crucial order parameter to drive isotropic-to-anisotropic structural transitions of chiral polar skyrmions to non-chiral merons, ...”, “we report the observation of a topological phase transition sequence in polar skyrmions, whereby tuning the temperature and strain boundary conditions, the anisotropic stripe phase deforms into an isotropic disordered circular phase, and finally transforms into an anisotropic ordered phase.” How to understand anisotropic, isotropic and isotropic-to-anisotropic structural transitions?

The structural transitions of polar textures and their (an)isotropy in 2D arrangements can be seen in the fast Fourier transform (FFT) patterns of the corresponding morphologies at different temperatures (insets, Fig. 3a-c). In the low-temperature labyrinthine phase which exhibits long stripes, it is anisotropic because there are two distinct peaks in the FFT along the pseudocubic $\pm(100)_{pc}$ directions. Similarly, the high-temperature phase is also anisotropic since the FFT shows four distinct peaks in $\pm(100)_{pc}$ and $\pm(010)_{pc}$ directions. In contrast, the room temperature phase which exhibits disordered bubble-like features, which corresponds to a ring feature in the FFT pattern. This means within the field-of-view of ~ 200 nm (Fig. 3b), there is no preferred orientational order in the room temperature phase, hence the term isotropic. This is detailed on page 7, lines 10-17 in Main Text.

Please clarify more about the way of stress loading or introduction in the process of changing temperature.

As detailed in page 13, lines 16-20 in the Main Text, the strain induced by changing temperature arises from the difference in thermal expansion coefficients of the SiN_x TEM grid and the oxide membrane (PbTiO_3 and SrTiO_3). The SiN_x grid consists of patterned circular holes of ~ 2 μm in diameter which serves as the viewing window and freestanding oxide membrane for TEM observations. The SiN_x and the oxide membrane are strongly glued together by van der Waals interactions, thus the edge of the circular hole acts as a “mechanical clamp” to the membrane. Therefore, upon heating (cooling) the oxide membrane would expand (contract) $\sim 5\times$ faster than the SiN_x , which results as the membrane being subjective to compressive (tensile) strain under the mechanical boundary condition. We note that the observed anisotropy at low-temperature phase can be explained by the inhomogeneous strain fields due to buckling of the oxide membrane.

7. The 4D-STEM method for describing the skyrmion polarization has been used in the previous Nature article (Das, S. et al. Nature 2019).

The method for polarization imaging used in this manuscript is a much-improved method (Kikuchi bands), which is more robust to crystal mis-tilt artifacts, compared to the one used in the Nature article (Bragg reflections). We note that the method employing polarity-sensitive Bragg reflections (used in the Nature paper) may be subject to serious artifacts, i.e., contrast reversal for mis-tilts as small as 0.04° , as shown in the new Fig S6D. This allows us to image much larger fields of view – instead of the single skyrmion in Fig 3D-G of the Das paper, we are able to map polarization for ensembles of hundreds of skyrmions and hence determine stacking arrangements. To illustrate this improvement in robustness, we simulated both

polarity sensitive Bragg reflections and Kikuchi bands in PbTiO_3 as a function of mis-tilt. We added this new Fig. S6 and attached as below:

Fig. S6. Diffraction effects and mis-tilt artifacts on the polarity determination of PbTiO_3 . Due to dynamical diffraction effects, the polarity-sensitive (A) Bragg reflections (e.g., $\pm(001)$, $\pm(002)$, and $\pm(003)$) and (B) Kikuchi bands (e.g., $\pm(001)$) along the polar axis exhibit intensity asymmetry. However, this intensity asymmetry can be largely affected by sample mis-tilt artifacts. Here, we simulate the CBED patterns of PbTiO_3 along $[100]_{\text{pc}}$ zone-axis with systematic crystal mis-tilt angles. (C) The intensity asymmetry in Bragg reflections can have contrast reversal with mis-tilts as small as 0.04° , while the Kikuchi bands in (D) can tolerate mis-tilt angles up to 0.23° . The diffraction patterns were simulated using 300 keV electrons with probe semi-convergence angle of 2 mrad, and PbTiO_3 thickness of 101 nm as consistent with the superlattice membrane. The intensity asymmetry is normalized as $I_{\text{asymmetry}} = (I_{hkl} - I_{\bar{h}\bar{k}\bar{l}}) / (I_{hkl} + I_{\bar{h}\bar{k}\bar{l}})$

The Figure 1 in the Nature paper shows that the three-layer sample is a striped structure, and the superlattice sample shows random skyrmion domains. However, the FFT image in Fig. 3C in the present manuscript is similar to RSM image in Figure 1 and Extended Data Fig.1 in the Nature paper. Please clarify them, and what is the main reason for the transition from strip-shaped elongated features to circular-shaped circular features, the temperature or the number of layers?

We note that for trilayer samples, the stripes (which are two-fold symmetric locally) can have two degenerate orientations in the bulk sample (e.g. Fig 1e), along pseudocubic $(100)_{\text{pc}}$ or $(010)_{\text{pc}}$, which gives rise to four peaks in the Q_x - Q_y cut of the x-ray RSM data. Although looking alike, this is different from the meron lattice, which is a superlattice under compressive strain shown in Fig. 3c, and has a genuine local four-fold symmetry, as is clear from the real-space image.

At different numbers of $\text{PbTiO}_3/\text{SrTiO}_3$ layers, the transition from stripe features to circular features may be explained by a change in the electrostatic boundary conditions. From trilayer to superlattice, the increase in number of superlattice periodicities led to the decreased depolarization field since the PTO to STO ratio is 1:2 for the trilayer while it is close to 1:1 for the superlattice. The effects of electrostatic energy on the vortex-to-skyrmion transition are discussed in the paper by L. Zhou *et al.*, *Matter* 5, 1031 (2022).

Figure S1 also shows that one is a long strip and the other is a circle. What is the mechanism of this different shapes? It is recommended to supplement the temperature-variable experiment in the three-layer structure.

We note that two different samples are imaged in Fig S1. Fig S1B is the trilayer and Fig S1C is the superlattice sample.

8. Finally, in the Abstract, the description “This topological phase transition is accompanied by a change in chirality, from net-handedness (in skyrmionic phase) to zero-net chirality (in meronic phase).” looks a little inconsistent with the title “Emergent chirality in a polar meron to skyrmion phase transition”.

We acknowledge the reviewer’s comment. We have modified the abstract so that it aligns with the title, which describes the meron to skyrmion direction of the transition. Going from Meron to Skyrmion produces chirality. Going from Skyrmion to Meron removes it. Both transitions are accessible.

Reviewer #3 (Remarks to the Author):

In this paper, the authors employed state-of-the-art experiment and simulation approaches to investigate the nontrivial topological polar textures and the associated phase transition in the oxide superlattice thin films. In the lift-off PTO/STO superlattices, they employed 4D-STEM to detect skyrmions and melons structures and developed new analysis method to determine the chiralities of each phases in experiments for the first time. In addition, they also reported the observation of topological phase transition sequences, whereby tuning the temperature and strain boundary conditions (induced by thermal expansion mismatch) conditions, the anisotropic stripe phase transformed to isotropic disordered circular phase, and finally transformed into anisotropic (square meron lattice). This is also the first experimental observation. Although some interesting questions were raised up instead of being answered in this work, it will certainly stimulate further research works to explore the macroscopic manipulation of the topological textures. This referee recommends the acceptance after the authors address one minor comment.

We thank the reviewer for the encouraging comments on the novelty both in characterization methods and the observed phenomena described in this manuscript.

The phase field simulation plays an important role in this manuscript to understand the transition (e.g., strain driven) observed in experiments. But the details are missing. What is the geometry of the simulation model? Is it bilayer materials, substrate bottom and oxide superlattice on top? In experiments, the thin film is placed over a pin-hole. The strain field could be non-uniform. It could be complicated if the z-direction relaxation was considered. Did the authors consider this in their modelling? Did they believe this detail is not significant for this study? Some discussion should be provided.

We thank this reviewer for the constructive comments and suggestions. We agree with the reviewer that “the phase field simulation plays an important role in this manuscript to understand the strain-driven transition observed in experiments” and we are happy to further clarify the details of the phase-field simulations as follows:

1. The simulation system is discretized into a three-dimensional box of 200*200*350, with each grid representing 1 unit cell.
2. As detailed in the supplementary information, we model the $[(\text{PbTiO}_3)_{16}/(\text{SrTiO}_3)_{16}]_8$ superlattice with substrate on bottom (30 unit cells) and air on top of the superlattice film (32 unit cells), similar to conditions used in paper by Das *et al.* (Nature 568, 368-372, 2019).
3. We agree that the strain can be rather non-uniform in experiments. However, we believe that the dominate effect is the global thermal-induced large compressive strain under heating. Experimentally, the phase observed near the edge is similar to that in the center of the 2- μm hole. Thus, for the simplicity of the model, we ignore the local strain inhomogeneity. More complicated model can be built which is out of the scope of this work.

The above details have been included in the Methods section (page 15, lines 5-17) in Main Text as well as in the Supplementary Text (page 3, lines 4-38).

Reviewers' Comments:

Reviewer #1:

Remarks to the Author:

The authors have addressed my comments well, i suggested its publications.

Reviewer #2:

Remarks to the Author:

I find that the authors have well addressed the raised concerns. Therefore, I am happy to recommend the paper for publication in Nature Communications.

Reviewer #3:

Remarks to the Author:

The authors have addressed all concerns from this referee.

Response Letter for Emergent Chirality in a Polar Meron to Skyrmion Transition

REVIEWERS' COMMENTS

Reviewer #1 (Remarks to the Author):

The authors have addressed my comments well, i suggested its publications.

We appreciate the referee's recommendations for publication in Nature Communications.

Reviewer #2 (Remarks to the Author):

I find that the authors have well addressed the raised concerns. Therefore, I am happy to recommend the paper for publication in Nature Communications.

We appreciate the referee's recommendations for publication in Nature Communications.

Reviewer #3 (Remarks to the Author):

The authors have addressed all concerns from this referee.

We appreciate the referee's recommendations for publication in Nature Communications.